# How do farmland rental markets affect farmers' income? Evidence from a matched renting-in and renting-out household survey in Northeast China

**Ning Geng**[1]*, **Zhifeng Gao**[2], **Chuchu Sun**[1], **Mengyao Wang**[1]

**1** School of Public Administration, Shandong Normal University, Jinan, Shandong, China, **2** Food and Resource Economics Department, University of Florida, Gainesville, Florida, United States of America

* gengning@sdnu.edu.cn

**Data Availability Statement:** All relevant data are within the manuscript and its Supporting information files. The data underlying the results presented in the study are all available.

## Abstract

Promoting farmland transfer through the farmland rental market is an essential instrument to achieve the scale economy of agricultural production in China. However, past literature on the land reform in China pays more attention to the renting-in household or the renting-out household, respectively, less to both types of households together. Using a large-scale survey of farm households in China, we examine the determinants of participation in the farmland rental market and quantify the impact of the rental market on farmers' income. Findings show household off-farm income, family members' part-time employment, agricultural subsidies, and participation in agricultural cooperatives significantly affect farmers' participation in the farmland rental market. Participation in the farmland rental market significantly increases the income of renting-in households, while it decreases the income of renting-out households, which might result from the temporary lag effect of the land system reform.

## 1. Introduction

Land reforms implemented since the beginning of the 1980s have greatly stimulated China's rural land rental market. Since land is not privately owned in many countries, liberalizing rural land use rights has long been advocated as a good policy for agricultural economic growth and peasants' well-being improvement in some developing countries. Starting 2002, there was an increased incidence of land rental activities in China (Deininger & Jin, 2005; Shuyi Feng et al., 2010) [1,2]. In the meanwhile, more and more rural young labors are pouring into China's big cities to take up non-farm employment, accelerating the process of urbanization. Therefore, the development of the rural land market is gaining momentum (Huang et al., 2012) [3]. The contribution to the economic growth of a well-functioning land rental market has well been recognized in China. Recently, the Chinese government further promoted rural land markets through "subcontracting, leasing, exchanging, or swapping" land-use rights

**Funding:** This research was funded by the ministry of education and social science youth program, grant number 17YJC79004 (host: NG) , the national natural science foundation of China program, grant number 71803104 (host: NG), and Qingchuang Science and Technology Support Plan for Colleges and Universities in Shandong Province grant number 2019RWE009 (principal member: NG).

**Competing interests:** The authors have declared that no competing interests exist.

(Huang et al., 2012) [4]. The land policy motivates some farmers to rent more land to increase farm size and reduce rural land fragmentation.

Development economics postulate that institutions play an essential role in facilitating economic growth (North, 1990) [5]. Securing property rights are expected to be a necessary factor of an institutional environment contributing to development (de Soto, 2000) [6]. So the arguments over property rights have long been the root cause of land reform, more often because the vast majority of the rural population rely on the land to generate livelihoods (Deininger K, 2005; Feng S., 2008; Becerri J., 2010) [1,7,8]. In particular, the reform of rural land resources' property rights is at the center of rural development and urbanization in contemporary China. As land is an essential factor of agricultural production, securing property rights is expected to not only improve productivity (Carol et al., 2015) [9] but also motivate investment in agricultural production (Jerzy Michalek et al., 2014) [10].

Land tenure security is an important precondition to developing a well-functioning rural land rental market (Besley, 1995) [11]. When rural land is rare, the certification of land use rights is expected to be a crucial way to impart greater security (Wang H., 2015) [4]. The land-use rights certification can identify the content of land use rights as well as confirm and secure long-term land tenure. Some studies show that enhanced tenure and land use right certification promote participation in the land rental market (Jinming Y.,2021) [12]. Findings in China show that tenure-enhanced land reforms increased the land market activities by 5.5% from the renting-in households (Deininger et al., 2015) [13]. Studies also show that land certification can positively affect renting-out households but leave renting-in households unaffected (Deininger et al., 2011) [14]. Land certification induces greater engagement in India's land rental market (Bardhan et al., 2014; Holden et al., 2011) [15,16] and appears to have stimulated more land rental market activities in the SNNP region in Ethiopia (Holden et al., 2016) [17]. Therefore improving tenure security and land certification is critical to the development of the land rental market.

As rural land rental market activity has increased in recent decades, so have the number of studies estimating its effects on farmers. Regarding the impacts of the rural land rental market, attention is mainly focused on efficiency, equity, and welfare. It is well known that the land rental market could have enhanced the allocative efficiency and agricultural productivity in rural China (Carter and Yao, 2002; Jin and Deininger 2009; Kimura et al., 2011) [18–20]. Research finds that land transfers from less efficient to more efficient farmers through land markets will help reallocate land. Empirical evidence from southeastern China and Ethiopia suggested that the rural land rental market could significantly boost economic growth and productivity because renting-in households have a higher marginal product of rural land than renting-out households (Shuyi Feng, 2010; Klaus Deininger, 2013) [2,21]. Regarding the effect on equity, several recent studies concluded that rural land rental markets could promote equity in some developing countries by transferring land from land-rich farmers to land-poor farmers and from labor-poor households to labor-rich household (Jacob Ricker-Gilbert, 2019; Chamberlin & Ricker-Gilbert, 2016; Yamano et al., 2009) [22–24]. Min Shi (2017) suggests that China's land rental market is a more effective method of rural land reallocation than administrative distribution without necessarily jeopardizing equity [25]. Besides, the welfare effect is primarily concerned with increasing income of renting-in or renting-out land and reducing poverty. The land rental market can facilitate greater access to land for small farms as a primary productive asset (Songqing Jin & T.S. Jayne 2013) [26]. The empirical evidence from dynamic panel income models shows that the small farms' total income gains from renting-in land were very remarkable (Klaus Deininger et al., 2013; Holden, Otsuka, and Place 2009) [21,27]. Renting-in land can increase the income of households and reduce the incidence of family poverty (Chen & Zhai, 2015) [28] The findings from Kenya show that renting-in land

would reduce poverty status by increasing per capita total income by 6.6% (Kimura S., 2011) [20]. Besides, off-farm employment also enables renting-out households to increase their earnings from the land rental and non-agricultural incomes (Feng & Heerink, 2008) [29].

Despite the general finding of positive effects from the rural land market, questions still remain about how the rental market improve income in the smallholder farming system for both renting-in and renting-out household. One of the major gaps in the literature is that most researchers focus on the land rental market's income effect either for renting-in households or for renting-out households. Few have examined the effect of the land rental market on both renting-in and renting-out households at the same time. An exception is a study by Chen & Zhai (2015), which found that the net income effect for renting-in household mainly originates in the scale expansion of farmland and improvement of technology efficiency, while the income effect of renting-out household is mainly from off-farm income increase and partly from rent revenue [28].

In this study, we focus on the rental land market in the Shandong province that lies on the coast of northeast China and uses recently collected data on renting-in household and their matched renting-out household to estimate the income effect from land renting. Shandong is a noteworthy case in several aspects. First, Shandong is a province with better economic development where rapid changes in land transformation have taken place in the context of industrialization and urbanization. Particularly, the province's superior geographical location provides many opportunities for farmers to engage in off-farm employment. Second, the province is a region of low flat land and an important grain production base. Thus, a well-functioning rental land market is crucial to ensure the nation's grain supply. Third, Shandong province is representative with a high incidence of land transactions. Because of the above reasons, it will be interesting to estimate the income effect and the determinants of rural land rental market participation in Shandong province.

Our research is motivated by the two questions: (1) would the improving welfare attributed to land rental markets in the previous studies hold if renting-in and renting-out households are observed equally in the same dataset? (2) What are the determinants of farmers' participation in the land rental market?.

We contribute to the current literature mainly from three aspects. First, we overcome the problem of estimating the income effect of renting-in and renting-out households separately. By compiling a dataset with renting-in households and their corresponding renting-out households, we consider both the demand and supply sides of the land rental market. Estimating the economic benefit to renting-in and renting-out households from land renting simultaneously, we can draw some conclusions that help make accurate land rental policies. Second, we use the propensity score method (PSM) to separate the effect of land renting from other factors that may affect the farm income. Thus the calculated results are relatively more convincing. Third, we investigate the determinants that lead to farmers' decisions on renting in or out rural land. Specifically, we determine whether the factors that affect households' decision to rent in or out are different and how the same factor may affect the land rental market participation in different ways.

The rest of this article is organized as follows. In Section 2, we develop a conceptual framework to describe the determinants of participation in the land rental market. Section 3 mainly introduces the study site, the procedure of data collection, and descriptive statistics. Section 4 describes the empirical model to analyze the possibility that farmers will rent in or rent out land and the PSM method to measure the income effect of land rental market. In section 5, we discuss the estimation results of the models. Section 6 shows the summary and conclusions of our study.

## 2. Conceptual framework

Following an agricultural household model of production and rural land market participation (Deininger and Jin, 2005; Jin and Deininger, 2009) [1,19], with the premise of "rational economic man", we develop a conceptual model to determine the factors affecting a peasant's participation in the rural land rental market.

Assume the household $i$ choose the rural land to farm (Ai), the allocation of household labors engaged in the farming ($L_{ia}$), the amount of labors for off-farm employment ($L_{io}$). We can use the following equation to model a household's income maximization behavior:

$$\text{Max } Income_i = pf(a_i, L_{ia}, A_i) + wL_{io} - R_i^{in}[(A_i - \overline{A_i})(r + MC_i^{in})]$$
$$+ R_i^{out}[(\overline{A_i} - A_i)(r - MC_i^{out})] \tag{2.1}$$

$$s.t. \ L_{ia} + L_{io} \leq \overline{L_i} \tag{2.2}$$

In this equation, $\overline{L_i}$ and $\overline{A_i}$ are the fixed number of rural labor and land endowments, respectively, of the $i$ household, while $a_i$ is a set of household characteristics; $f(a_i, L_{ia}, A_i)$ is an agricultural production function; $p$ stands for the price of agricultural products; $w$ denotes the wage rate of off-farm employment for $L_{io}$; and $r$ is a competitive rent in the rental land market. $R_i^{in}$ and $R_i^{out}$ are indicators standing for the renting-in ($R_i^{in} = 1$ for renting-in land or 0 otherwise) and the renting-out ($R_i^{out} = 1$ for renting-out land or 0 otherwise), respectively, of land. $MC_i^{in}$ and $MC_i^{out}$ are the respective transaction costs for renting land in the land rental market.

Solve the income maximization problem; the following two equations define the conditions of the renting-in and renting-out of land:

$$R_i^{in} = f\left(a_i, \overline{L_i}, L_{ia}, L_{io}, A_i, \overline{A_i}, MC_i^{in}, MC_i^{out}, w, r\right) \tag{2.3}$$

$$R_i^{out} = f\left(a_i, \overline{L_i}, L_{ia}, L_{io}, A_i, \overline{A_i}, MC_i^{in}, MC_i^{out}, w, r\right) \tag{2.4}$$

Suppose $r$ is different for the local households due to the different plots of land. $MC_i^{in}$ and $MC_i^{out}$ are consistent for all farmers because they may face the risk of land loss, and the two indicators are related to the land tenure security (or certification of land renting contract), thus we can incorporate $r$, $MC_i^{in}$, $MC_i^{out}$, $A_i$, and $\overline{A_i}$ into the vector $Mi$ which denotes rental market transaction variables. The labor variables ($\overline{L_i}, L_{ia}, L_{io}$) and $a_i$ can be represented by a vector of household characteristics ($Hi$), and $Di$ is assumed to be family off-farm employment. Besides, $w$ can be included in the external environment ($Wi$). Hence we can derive the two reduced-form function associated with the Eqs 2.3 and 2.4 as follows:

$$R_i^{in} = f\left(D_i, H_i, M_i, W_i\right) \tag{2.5}$$

$$R_i^{out} = f\left(D_i, H_i, M_i, W_i\right) \tag{2.6}$$

The Eqs (2.5) and (2.6) show that the determinants of renting-in or renting-out lands are determined by family off-farm employment ($D$), household characteristics ($H$), rental market transaction characteristics ($M$), as well as the external environment ($W$).

## 3. Survey and data

### 3.1 Study site

Shandong is an important agricultural province that lies on China's east coast and across the sea from Japan and South Korea (Fig 1), with a location suitable for an export-oriented economy. Shandong is also one of the most densely populated provinces of China, with small farms and high levels of land fragmentation cover approximately 157,100 km2. As the major grain-producing region, Shandong's grain production in 2017 was 53.2 million tons (Fig 2), accounting for 8 percent of the national total. Shandong led the country in total grain output value in 2017 (Fig 3), reaching $60.635 billion. In sum, Shandong is a typical region that characterizes the plain area of northeast China and also a production base for ensuring food security.

Besides, Shandong has about 24.2 million rural laborers working off-farm by 2017. An efficient rural land market would not only help improve land productivity but also increases rural income. The Chinese government has promoted the reform of rural land property rights since 2015. The innovation of the land reform lies in liberalizing the land usage (or management) right[1] (In China, rural land ownership belong to the rural collective organization, farmers only have contracted management and use rights.) and activating the land rental market according to supply and demand of farmland (Wang H, 2015) [30], which can realize the scale economy of farmland and improve peasants' income (Vandeplas et al, 2013) [31].

However, due to the influence of traditional Chinese thinking, in Shandong, many small-scale landholders are not willing to rent out their farmland, although these smallholders operate their land very inefficiently. In the meanwhile, productive farmers may not have the

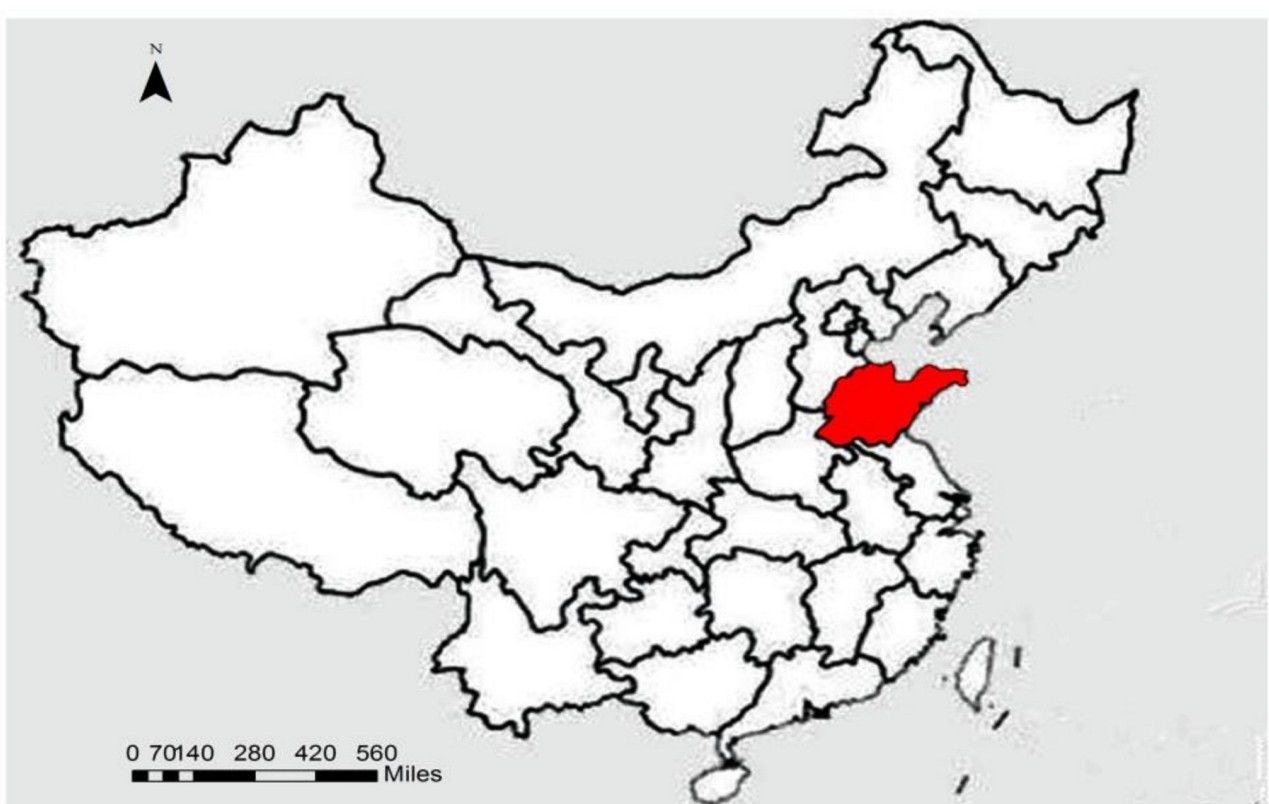

**Fig 1. A map of the study area and sample distribution.**

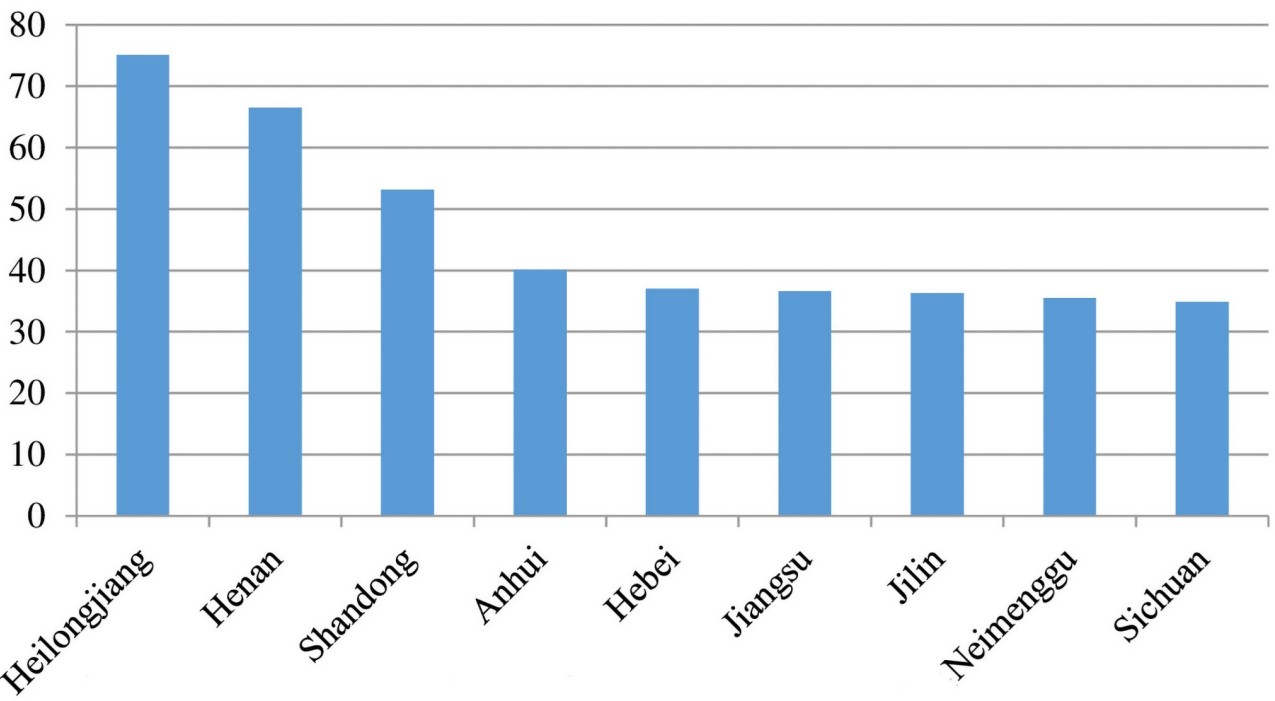

**Fig 2. Gross grain output of China's provinces (top 9) in 2007.** Unit: Million tons. Data source: China's National Bureau of Statistics (CNBS) 2007.

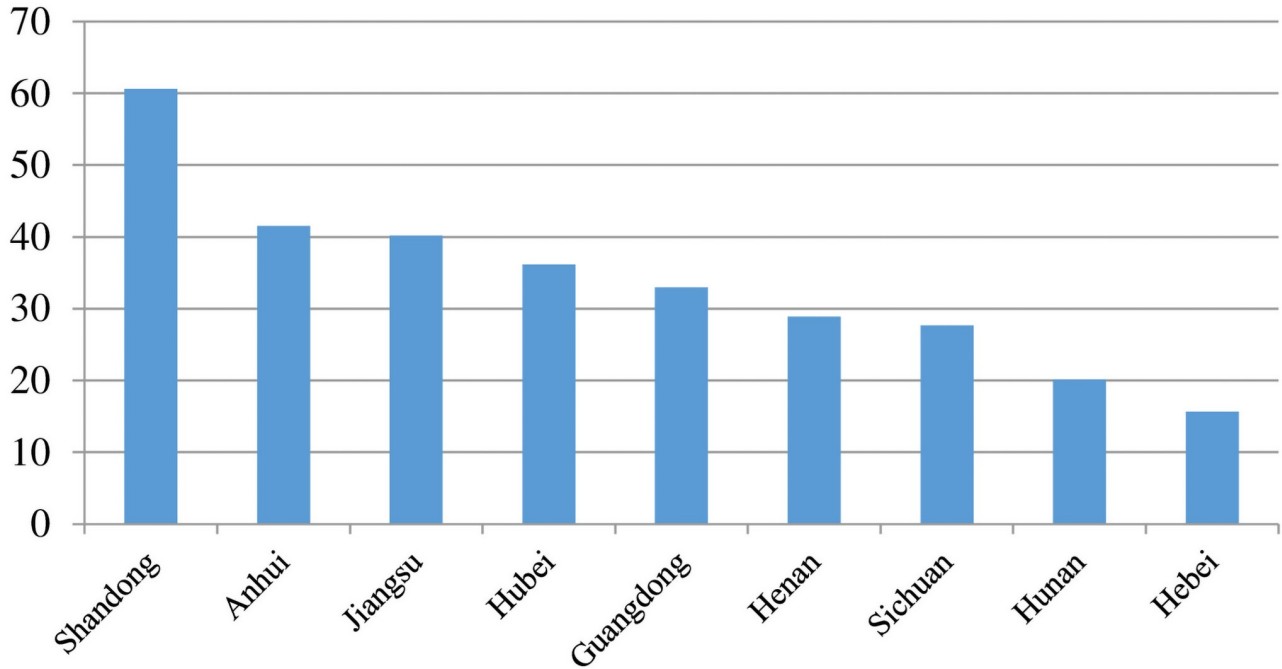

**Fig 3. Gross grain output value of China's provinces (top 9) in 2007.** Unit: Billion dollars. Data source: China's National Bureau of Statistics (CNBS) 2007.

**Table 1. Definition of relevant indicators of households' livelihoods.**

| Category | Definition | Subitem | Abbreviation |
|---|---|---|---|
| $H_i$ | householder characteristics | 1.the age of householder | *age* |
| | | 2.the education of householder | *edu* |
| $D_i$ | family off-farm employment | 1.the share of household off-farm income | *share* |
| | | 2. the ratio of family members'off-farm employment to farm employment | *parttime* |
| $M_i$ | the land rental market transaction | 1.the operating area of land | *area* |
| | | 2.a long-term certificate for land tenure | *cont* |
| | | 3.the price of renting land | *price* |
| | | 4. the subsidies for production | *subsidy* |
| $W_i$ | the external environment | 1. the distance from the village to the urban area | *distance* |
| | | 2. the agricultural cooperative | *cooper* |
| | | 3.agricultural technical training family members have received | *tech* |

experience to manage large-scale farmlands when they make a decision on renting in more land. Therefore, it is essential to determine the factors influencing farmers' participation in the land rental market and whether participation in the land rental market can improve rural income.

## 3.2 Data collection

The data used in this article were collected from a socio-economic survey of landholders from December 2017 to November 2018 during the non-farming season in Shandong. The survey collected detailed information on family members' socioeconomic characteristics, such as household income, on-farm and off-farm activities, the amount of renting land, land-use history, and land certification status.

To ensure that the sample includes typical renting-in and renting-off households, sample selection was carried out as follows. First, the survey was conducted in two cities of Shandong province, Dezhou and Qingdao. Both regions are plain areas with large plots and have an active land rental market in Shandong. More importantly, they are agricultural demonstration zones in Shandong, and more than 40% of farmland is traded in the land market. Specifically, the proportion of traded land in Qingdao is approximate 50%, while it is 42% in Dezhou. Second, because we wanted to interview both the land renting-in (renting-out) households and their corresponding renting-out (renting-in) households, the sample areas selected were relatively large villages with many rural land transactions. Thus, four administrative townships were randomly selected from Dezhou city, including three administrative villages in each sample township. While in Qingdao, two administrative townships were selected, and in each sample township, five administrative villages were selected.

Finally, to increase the response rate and obtain respondents' truthful opinions, we collected the data using a face-to-face survey. During the survey, one enumerator took 1–2 hours to complete one questionnaire, and each survey enumerator completed 3–5 questionnaires per day. To ensure the authenticity of the survey, we randomly selected respondents in the sample villages. Because our work focused on the income effect of the land rental market on land renting-in households and their corresponding renting-out households, when we interviewed the renting-out households, we tried to identify their matching renting-in households by the land lease contract. To estimate the land rental market's income effect, we also interviewed farmers who were not participating in the land rental market in the sample villages, which was used as a control group.

With the data covering 810 rural households across 22 villages from Shandong province, our study can be considered a representative sample of the rural land rental market participants. There are 407 participating households, including 240 renting-out households and 167 renting-in households. Moreover, most of the land lease agreement was between 8–20 years. Because the current study mainly focuses on the welfare effect of participation in the land market, we did not examine the impact of lease agreement length on the land rental market participants' income. Future research can investigate the effect of land rent-in/rent-out contract length on household welfare.

## 3.3 Definition and descriptive statistics

In this section, we define and describe the critical variable in the model that may affect farmers' participation in the land rental market and their household income.

As discussed in Section 2, land rent in and rent out behaviors are influenced by four sets of variables such as household characteristics ($H_i$), family off-farm employment ($Di$), rental market transaction characteristics ($M_i$), as well as the external environment ($W_i$) (Eqs 2.3 and 2.4). The four dimensions and their relevant indicators (Table 1) are discussed below.

$H_i$ denotes a vector of household characteristics including age ("age") and education ("edu") of the householder. Generally, the average age of the renting-in group is significantly lower than that of the renting-out group because young households are more willing to rent in farmland for scale farming (Min et al, 2015) [32]. In addition, the renting-out farmers have a significantly higher education level because well-educated farmers have more opportunities to find off-farm jobs.

$D_i$ is a vector of variables representing the number of farms and off-farm labors. In our study, we use the share of household off-farm income ("share") and the ratio of family members' off-farm employment to farm employment ("parttime") as proxy variables of $Di$. As the rapid development of urbanization, the non-agricultural activities of ordinary farmers are gradually transferred to the service industries in the surrounding areas or urban areas. Therefore, the higher the proportion of non-agricultural income or the ratio of family members' off-farm employment, the more inclined farmers are to rent out more farmland. In today's China, renting a lot of land requires hiring labor, and hiring labor can replace family members.

$M_i$ includes the rental market transaction variables, such as land operating area (area), a long-term certificate for land tenure (cont), the price of renting land (price), and the subsidies for production (subsidy). The land operating area is an important factor affecting rural land transfer as farmers with more land are more likely to rent land out (Huang and Ji, 2012) [3]. Additionally, the land price (e.g., land rent) is a key element in the land rental market (Wang et al., 2011b) [33]. A high price of renting land may motivate more farmers to rent the land out. Besides, a long-term land tenure certificate is helpful to the development of the land renting market. It has been promulgated to peasants for ensuring land use right with a stable contract period since the "Rural Land Contract Law" was issued in China in 2002 (Huang and Ji, 2012; Deininger et al., 2014) [3,34]. At last, the subsidies for production (subsidy) will induce more demand for renting-in land because they can reduce the production cost when farm operation is expanded to a large production scale (Chen.F,2015) [28].

$W_i$ is the fourth dimension denoting the external environment that may affect the land transfer. The variables included are the distance from the village to the urban area ("distance"), which can be used to represent the off-farm opportunity information of household members. Furthermore, participation in agricultural cooperatives could promote scale economies or reduce transaction costs to some extent (Tafesse W. et al., 2019) [35]. So the control variable "cooper" could have a significant impact on the renting-out and renting-in land. At present, in

**Table 2. Descriptive statistics of variables in the model.**

| Variables | Description | non-participants | rent-out group | rent-in group | Diff B-A | Diff C-A |
|---|---|---|---|---|---|---|
| | | Mean A | Mean B | Mean C | T | T |
| **income** | family per capita net income (Yuan) | 7718.87 | 7075.54 | 73418.1 | 0.94 | -11.79*** |
| **age** | age | 59.05 | 56.55 | 47.27 | 2.66*** | 12.31*** |
| **edu** | education | 1.66 | 1.93 | 2.54 | -4.67*** | -13.97*** |
| **share** | the share of household non-farm income | 0.51 | 0.42 | 0.17 | 2.91*** | 10.19*** |
| **parttime** | the number of off-farm employment | 1.39 | 1.85 | 2.16 | -7.16*** | -13.86*** |
| **area** | the operating aera of land (mu) | 5.84 | 10.17 | 189.84 | -7.37*** | -17.74*** |
| **price** | land rent in rental market (Yuan per mu) | 1276.39 | 1420.38 | 768.42 | 1.34 | 4.27*** |
| **cont** | = 1, if have a land contract; = 2, no | 1.13 | 1.20 | 1.00 | -2.54** | 4.97*** |
| **subsidy** | = 1, if receive subsidies; = 2, no | 1.01 | 1.30 | 1.00 | -8.54*** | 1.45 |
| **tech** | = 1, if gaining technical training; = 2,no | 1.96 | 1.82 | 1.02 | 6.08*** | 57.03*** |
| **cooper** | = 1, if join an cooperative; = 2, no | 1.95 | 1.68 | 1.77 | 9.82*** | 6.66*** |
| **distance** | the distance from the village to the city | 33.1 | 27.51 | 29.14 | 9.34*** | 5.74*** |

Note:

*, **, and *** denote the significant coefficients at the levels of 10%,5%, and 1%, respectively.

The differences in the means between groups were tested using the student T-test.

China, agricultural production has shifted from labor-intensive to technology-intensive production, so whether family members have received special technical training represents farmers' technical level. The "tech" variable indicates farmers' knowledge of agricultural technologies. More knowledgeable farmers are more likely to rent in the land to develop efficient agriculture.

The statistical descriptions of all variables used in the regression are summarized in Table 2. The statistics in Table 2 show that the renting-in group's income level was significantly higher than that of those not participating at 1% significance level. The income level of the renting-out group was the lowest of the three groups. However, the statistical difference of the above indicators may not be the inevitable result of land renting, but caused by other factors. Therefore, we need to establish other econometric models to test the impact of land renting on household income.

## 4. Empirical model

Following the above conceptual models, in this section, we will establish econometric models to estimate the determinants of farmers' renting-in and renting-out land and use the propensity score matching (PSM) method to measure the income effect of the land rental market.

### 4.1 Model specification: The Logistic model

The determinants of farmers' renting-in or renting-out decision on rural land can be modeled with the Stochastic Utility Decision Model (Ali, A., and Abdulai, A., 2010) [36]. We postulate $U1$ and $U0$ representing the utility of farmers' participation in the land rental market and non-participants, respectively. If $V^* = U1 - U0 > 0$, the farmers will choose to participate in the rental land market, otherwise not. Although we can't observe $V$, it can be represented as a function of observable variables. We specify the following function models:

$$V^* = f(X) + \mu, V = \begin{cases} 1 \; V^* > 0 \\ 0 \; V^* < 0 \end{cases} \qquad (4.1)$$

In function (4.1), $V$ is a two-value variable. If one household participates in the land rental market (renting in or renting out), $V$ is equal to 1, otherwise, $V$ is equal to 0. $X$ is the vector of exogenous explanatory variables affecting farmers' participation in the land rental market. The detailed definitions and statistical descriptions $X$ are summarized in Tables 1 and 2. $\mu$ is a random disturbance term.

We can estimate the determinants of farmers' renting-in and renting-out land decisions by Logistic model. Specifically, the probability that a farmer rent in or rent out her/his land is

$$P\left(Y=1|X\right)=F(X,\beta)=\varphi\left(X'\beta\right)=\frac{\exp\left(X'\beta\right)}{1+\exp\left(X'\beta\right)} \quad (4.2)$$

Suppose $P=P(Y=1|X)=\frac{exp(X'\beta)}{1+exp(X'\beta)}$, and $1-P=P(Y=0|X)=\frac{1}{1+exp\ (X'\beta)}$, we can get the following equation,

$$\frac{P}{1-P}=\exp\left(X'\beta\right) \quad (4.3)$$

Taking the log of both sides of Eq 4.3, we get another equation as follows,

$$\ln\left(\frac{P}{1-P}\right)=X'\beta \quad (4.4)$$

Thus, the sign of an independent variable's coefficient indicates whether changes in the independent variable would increase or decrease the probability that a farmer rents in or rents out the land.

## 4.2 Estimation approach: Welfare effect of land rental market

Previous studies have shown that whether to participate in the land rental market, farmers are "self-selecting", and some unobservable factors (such as production preference, management skills or family wealth, etc.) may affect the decision-making and lead to biased estimation results. For this purpose, Rubin (1974) put forward "a counterfactual framework", which was referred to as "Rubin Causal Model (RCM)" [37]. The dichotomous Di = {0, 1} indicates whether the household $i$th participates in the land rental market. Correspondingly, the counterfactual outcomes are $y_{1i}$ and $y_{0i}$, respectively, participation and non-participation in the land rental market.

$$y_i = \begin{cases} y_{1i}\ if\ V_i=1 \\ y_{0i}\ if\ V_i=0 \end{cases} \quad (4.5)$$

The treatment effect for $i$ is thereby ($y_{1i}$- $y_{0i}$), which is what we want to estimate. $y_{1i}$ and $y_{0i}$ can not be observed at the same time. So we can model the household $i$'s observed outcome as follows:

$$y_i\ =(1-V_i)y_{0i}+V_i\,y_{1i}=y_{0i}+V_i\left(y_{1i}-y_{0i}\right) \quad (4.6)$$

In Eq (4.5), $y_i$ is impacted by the household $i$'s traits set $X_i$, the exogenous explanatory variables set $X_j$, and the random unobservables $\mu_i$. For the same household $i$, $\mu_i$ represents the unobserved determinants of outcomes following a joint normal distribution, such as the household's production preferences, managerial skills, or household wealth. On this basis, the influence equation of participating in farmland capitalization on farmers' welfare can be set as

follows:

$$y_i = f_0\left(X_i, X_j\right) + \mu_{0i} + V_i\left[\left(f_1\left(X_i, X_j\right) + \mu_{1i}\right) - \left(f_0\left(X_i, X_j\right) + \mu_{0i}\right)\right]$$
$$= f_0\left(X_i, X_j\right) + V_i\left[f_1\left(X_i, X_j\right) - f_0\left(X_i, X_j\right) + \left(\mu_{1i} - \mu_{0i}\right)\right] + \mu_{0i} \quad (4.7)$$

Moreover, the household *i*th participation in the land rental market does not affect other individuals, which is deemed to "stable unit treatment value assumption" (SUTVA). This assumption excludes inter-individual social interactions or general equilibrium. Because the treatment effect ($y_{1i}$- $y_{0i}$) is a random variable, we just need to pay attention to the expected value E.

To solve "self-selection" of research objects and the influence of unobservable factors (such as production preference, management skills or family wealth, etc.), this article adopted the propensity score matching (PSM) method. We selected farmers who did not participate in the land rental market as the control group for the matching analysis. Rosenbaum & Rubin(1985) constructed a counterfactual analysis framework different from Eq (4.7) to effectively eliminate the biased estimation of non-random distribution of samples [38]. We need to find a control group similar to the treatment group to reduce the bias of sample selection. For those having participated in the land market, we estimate the average treatment-on-the-treated effect (ATT) by:

$$ATT = E(y_{1i}|X_i, X_j, D_i = 1) - E(y_{0i}|X_i, X_j, D_i = 1)$$
$$= E(y_{1i} - y_{0i}|X_i, X_j, D_i = 1) \quad (4.8)$$

Eq (4.8) shows that $E(y_{1i} \mid X_i, X_j, D_i = 1)$ is observable, while $E(y_{0i} \mid X_i, X_j, D_i = 1)$ is unobservable, which is called a counterfactual result. Therefore, an alternative indicator $E(y_{0i} \mid X_i, X_j, D_i = 1)$ can be constructed using the propensity score matching method (PSM). On the premise that the covariates $(X_i, X_j)$ of the two groups are as similar or identical as possible, the propensity score of each sample to enter the treatment group is calculated. This method can accurately evaluate the income effect of the land rental market and compare the source of the income effect of two groups of matched samples.

## 5. Estimation results and discussion

### 5.1 Determinants of farmers' participation in the land rental market

Before the regression estimation, we first analyze the correlation of the covariates variables $X$ in the model and test the possible multicollinearity problems. Table 3 shows that the average variance inflation factor (VIF) for the land rent-out and rent-in groups are 1.24 and 1.45, respectively. The largest VIF of the control variables for both groups is 2.58. Therefore, it can be concluded that the multicollinearity problem in the models is not serious.

To further measure the income effect of the land rental market on different households, and to provide conditional probabilistic fitting values for calculating the household income effects (ATT), a flexible logistic model was used for regression estimation. We first estimate two functions as Eqs (2.5) and (2.6). The regression results are reported in Table 4. The following conclusions can be drawn from the coefficient estimates of the key variables in the two models.

First, in the model for renting-in land, the coefficients of "age", "share", "parttime", "price", "subsidy", and "tech", are significant at the levels of 1% (Table 4). The results imply that relatively young households are more willing to rent in farmland for scale farming. The negative

**Table 3. Measurement results of variance inflation factor and tolerance of each variable.**

| Variable | rent-out | | rent-in | |
|---|---|---|---|---|
| | VIF | TOL | VIF | TOL |
| age | 1.4 | 0.72 | 1.53 | 0.65 |
| edu | 1.42 | 0.7 | 1.64 | 0.61 |
| share | 1.32 | 0.76 | 1.53 | 0.66 |
| parttime | 1.22 | 0.82 | 1.63 | 0.62 |
| area | 1.3 | 0.77 | 1.74 | 0.57 |
| price | 1.16 | 0.86 | 1.13 | 0.88 |
| cont | 1.14 | 0.87 | 1.08 | 0.92 |
| subsidy | 1.14 | 0.88 | 1.02 | 0.98 |
| tech | 1.3 | 0.77 | 2.58 | 0.39 |
| cooper | 1.11 | 0.89 | 1.27 | 0.79 |
| distance | 1.29 | 0.78 | 1.23 | 0.81 |
| Mean VIF | 1.24 | | 1.45 | |

Note: VIF: Variance inflation factor; TOL: Tolerance.

coefficient of "share" implies that the lower the proportion of non-agricultural income, the more likely is a household willing to rent in the land. And the "parttime" denotes the number of household members who have off-farm employment. Due to the low comparative returns of agriculture, ordinary farmers' non-agricultural activities are gradually transferred to the primary service industries in the surrounding areas and urban areas. Therefore, the households with more family members having off-farm jobs are more inclined to rent in farmland. These farmers can earn higher income from off-farm employment to hire laborers for farming

**Table 4. Results of Logistic model: Determinants of farmers' participation in land rental market.**

| variables | renting-out | | renting-in | |
|---|---|---|---|---|
| | Coefficient | Std.error | Coefficient | Std.error |
| Constant | 5.11 | 2.01 | 27.47*** | 8.424 |
| age | -0.001 | 0.012 | -0.269*** | 0.071 |
| edu | 0.418** | 0.191 | 0.261 | 0.917 |
| share | -1.388*** | 0.363 | -9.583*** | 2.762 |
| parttime | 1.311*** | 0.19 | 4.934*** | 1.285 |
| area | 0.084*** | 0.027 | | |
| price | 0.0003 | 0.0009 | -0.001*** | 0.003 |
| cont | 0.295 | 0.355 | | |
| subsidy | 2.518*** | 0.313 | 4.135*** | 0.232 |
| tech | -0.334 | -0.498 | -6.741*** | 1.289 |
| cooper | -4.051*** | 0.478 | -4.316* | 2.619 |
| distance | -0.109*** | 0.019 | -0.044 | 0.065 |
| R-squared | 0.441 | | 0.92 | |
| Log likelihood | -237.39 | | -25.26 | |
| Number of oservations | 641 | | 570 | |

Note:

*, **, and *** denote the significant coefficients at the levels of 10%,5%, and 1%, respectively.

(Deininger and Jin, 2005; Chen & Zhai, 2015; Jacob Ricker-Gilbert, 2019) [1,23,28]. Table 4 also shows that the other explanatory variables, e.g., the price of renting land, the agricultural cooperative, and agricultural technical training, significantly affect farmers' participation in the land rental market. In line with our theoretical analysis, land price is a key element of the land rental market (Jerzy Michalek et al., 2014) [10]. The higher the land rent is, the less likely a household is to rent in the land (Carol et al., 2015) [9]. The "tech" variable clearly indicates that the households receiving technical training are more likely to rent in the land. At present, agricultural production has shifted from labor-intensive to technology-intensive production, so farmers' technical skills related to efficient agricultural production technologies should significantly affect their land renting behavior (Carter and Yao, 2002; Jin and Deininger 2009; Kimura et al., 2011; Jerzy Michalek et al., 2014) [10,18–20]. The control variable "cooper" also indicates that households who are members of the agricultural cooperatives are more likely to rent in farmland. Joining cooperatives may reduce transaction costs, thus increase households' likelihood to rent in the land.

Second, in the model for renting-out land, the positive coefficient of "edu" indicates that a higher level of landholders' education will increase their likelihood of renting out the land because these people have higher chances to find off-farm jobs (Yamano et al., 2009; Shuyi Feng, 2010; Klaus Deininger, 2013) [2,21,24]. Moreover, the variables "share", "parttime", "subsidy", and "cooper" have the same impact on renting out land as in the model for renting-in land. This indicates that the above factors influence the participation of farmers in the land rental market (Feng & Heerink, 2008; Songqing Jin & T.S. Jayne 2013) [26,29]. Furthermore, the land area (area) is positively correlated with households' land renting-out behavior. Households with more land are more likely to rent out the land. This might be because land-rich households have more chances to earn money instead of specializing in farming (Klaus Deininger et al., 2013) [21]. Finally, the distance from the village to the urban area is significantly negatively associated with the probability that a household rents out the land. The distance to the urban area can be used as an indicator of the cost for farmers to go out for work. The further it is to the urban area, the fewer off-farm employment opportunities, and farmers are less willing to rent out their land (Macmillam, 2000; Dijk, 2003) [39,40].

## 5.2 Measurement and analysis on income effect of the land rental market

The most important purpose of propensity score estimation is to balance the distribution of variables between experimental and treatment groups rather than to obtain accurate probabilistic predictions. Figs 4 and 5 indicate the region of common support of transfer households (renting-out households and renting-in households) and non-participants. Compared with the total sample size, the sample loss ratio is small, and the condition of the common support domain is satisfactory.

Moreover, to minimize the inter-group differences in the probability distribution of sample households of different groups and ensure high-quality matching results, we carried out a balance test on explanatory variables and control variables of the experimental and treatment groups. Table 5 shows the difference before and after the match. The bias (in %) of explanatory and most control variables after the matching have shrunk to less than 10%, and most of the T-test result is significant after the match. Especially important, matching can not only reduce the total bias but also can effectively reduce the differences between groups of different samples. The standardized deviation of individual control variables is more than 10% after matching but still within the acceptable range. The above test results show that the matching of explanatory variables and control variables between different samples is desirable. It has certain persuasive power to explain the change of farmers' welfare level.

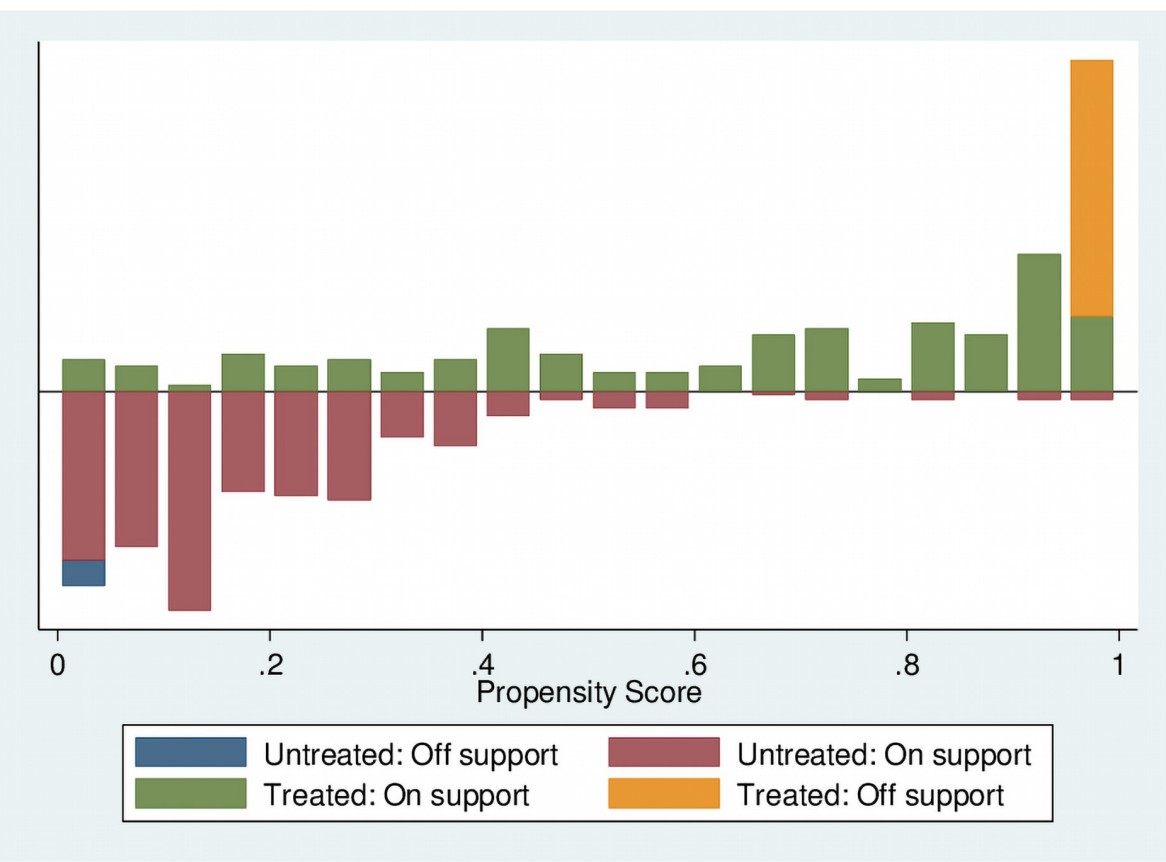

**Fig 4. After matching of renting-out households.**

After obtaining effective matching samples, we calculated the income effect of the land rental market (ATT) as in Eq (4.8). Stata12.0 software was used for the actual matching. To reduce the loss of effective samples, the matching was carried out with a replacement, and different matching methods were used because the selected sample individuals (control group) in this study were limited. The results in Table 6 show that, from a quantitative perspective, although the values of the four matching results are different, the numerical directions are consistent, and most of them have passed the significance test. Based on the comparative analysis of the two main welfare effects, the welfare effect of renting-out farmland is negative while renting-in farmland has a significant increase in income. In other words, the average per capita net income of a renting-out household will decrease by 567.29 yuan. In comparison, the average per capita net income of a renting-in household will increase by 26386.53 yuan.

Considering the limitations of PSM, the implicit bias caused by unmeasurable factors should be considered. That is, the influence of the implicit bias on the matching results should be tested. We used the "Rosenbaum Bounds" rule to test the sensitivity of the matching results (Rosenbaum,1985) [38]. Test results show that when $\Gamma = 1.5$, the corresponding P-value is still significant at 5% or 10% levels. This indicates that the effect of unobserved confounding on the ATT estimates is relatively small, and our conclusion on the welfare effect of the land renting market is reliable.

Based on the above regression results (Table 4) and the calculated results of the income effect of farmers (Table 6), the following conclusions can be made. Firstly, the land rental

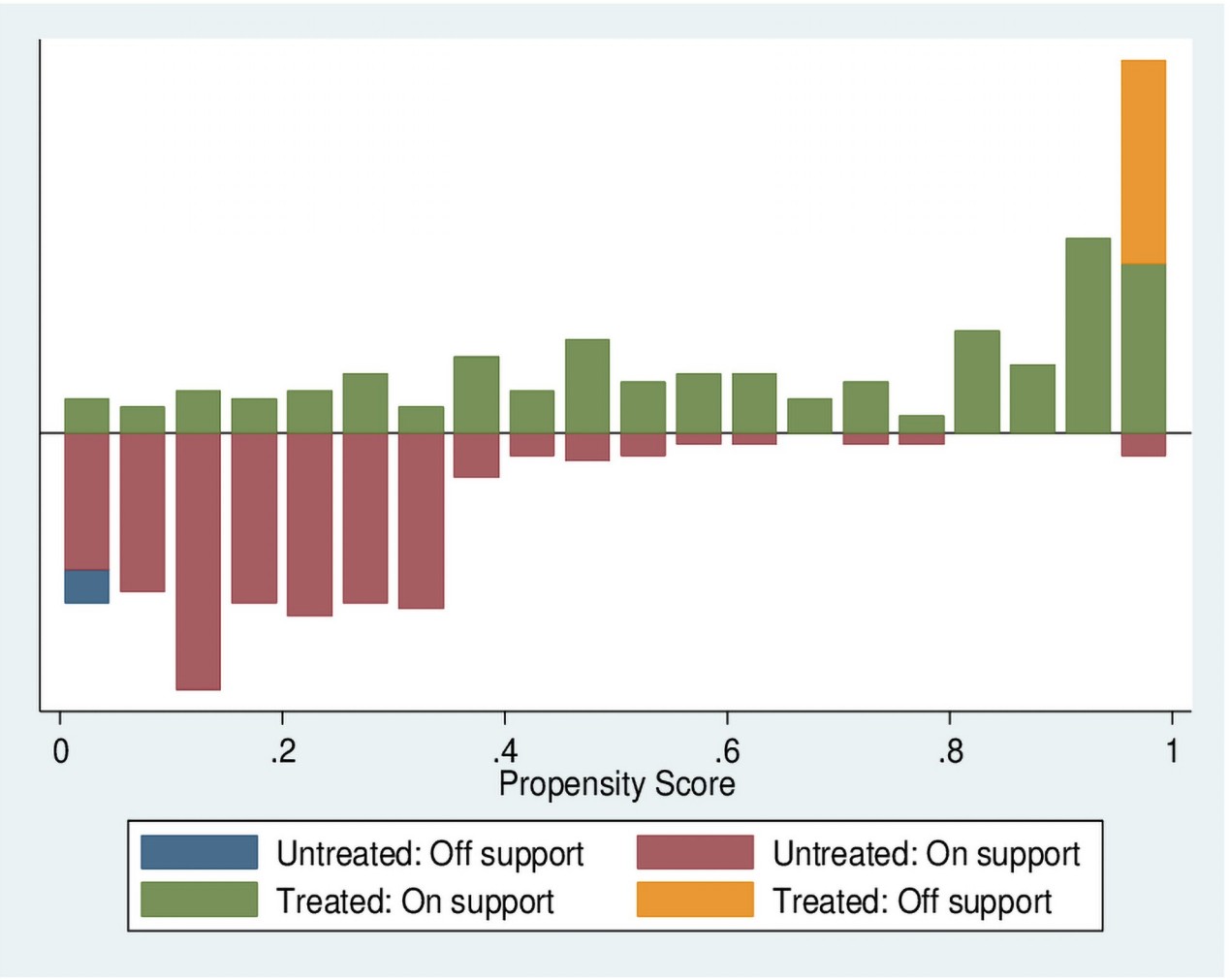

**Fig 5. After matching of renting-in households.**

market can significantly increase the income level of the renting-in household. This may mainly relate to the government's financial subsidies for production. However, the income level of renting-out households is declining.

Secondly, for the renting-out household, "education", "the share of household non-farm income (share)", "the ratio of family members'part-time employment (parttime)", "household contracted farmland area (area)", "whether to join agricultural cooperatives (copper)", and "the distance from the village to the city (distance)" are the key elements for the renting-out households. Theoretically speaking, farmers are willing to voluntarily rent out their land only when the total income from the land rent and non-agricultural income is higher than the net income of agricultural production. However, the income level of renting-out households has not been improved significantly, which may arise due to the lag effect of the land system reform (Besley and Burgess 2000) [41].

Thirdly, for the renting-in households, The results show that reducing production cost ("subsidy") and improving technology level ("tech") are the key influencing factors of current land scale management households. The variables of "age" indicate the ability of farmers to operate and manage the land. Meanwhile, farmers' scientific and technological levels and the

**Table 5. Balance test of control variables by nearest neighbor matching method.**

| variables | unmatched | renting -out | non-participants | bias % | renting -in | non-participants | bias % |
|---|---|---|---|---|---|---|---|
| | matched | Mean | | | Mean | | |
| age | U | 56.93 | 58.91 | -17.5 | 52.21 | 58.91 | -31.2 |
| | M | 56.84 | 56.61 | 1.6 | 53.42 | 56.61 | 6.4 |
| edu | U | 1.93 | 1.67 | 33.2 | 2.54 | 1.68 | 122.2 |
| | M | 1.906 | 2.0573 | -19.1 | 2.33 | 2.36 | -5.1 |
| share | U | 0.48 | 0.53 | 14.6 | 0.32 | 0.53 | -53.8 |
| | M | 0.52 | 0.47 | 13.5 | 0.36 | 0.47 | -27.6 |
| parttime | U | 1.88 | 1.43 | 11.7 | 2.01 | 1.43 | 2.01 |
| | M | 1.74 | 1.78 | -3.9 | 1.78 | 1.78 | 1.78 |
| area | U | 9.57 | 6.05 | 44 | 89.84 | 6.05 | 75.6 |
| | M | 7.25 | 9.54 | -28.6 | 54.97 | 9.54 | 40 |
| tech | U | 1.84 | 1.96 | 13.9 | 1.44 | 1.96 | -37.8 |
| | M | 1.89 | 1.74 | 6.5 | 1.52 | 1.74 | 24.9 |
| price | U | 1447.2 | 1366.3 | 5.6 | 1115.8 | 1366.3 | -17.1 |
| | M | 1536.6 | 1677.7 | -9.8 | 1204.8 | 1677.7 | -12.4 |
| cont | U | 1.24 | 1.12 | 30.2 | 1.12 | 1.12 | 14.4 |
| | M | 1.16 | 1.12 | 10.9 | 1.09 | 1.12 | -2.5 |
| cooper | U | 1.71 | 1.93 | -61.5 | 1.74 | 1.93 | -56 |
| | M | 1.76 | 1.98 | -60.4 | 1.88 | 1.98 | -9.1 |
| distance | U | 26.79 | 32.75 | -91.2 | 29.14 | 33.15 | -53.3 |
| | M | 27.38 | 32.11 | -72.4 | 28.4 | 28.89 | -6.5 |

agricultural subsidies determine the output boundary of the farmland and the profit space of scale operations.

## 6. Conclusion

This study evaluates the income effect of farmers participating in the land rental market and analyzes farmers' behavior relevant to participation in the land rental market. Taking account of four dimensions of livelihoods decision-making of farmers, we find that the land rental market increased the level of income for farmers who rented in farmland while reducing the level of income for farmers who rented out their land. Two reasons may contribute to the result. Firstly, given the long-term goal of gaining economies of scale in agriculture, up-to-bottom

**Table 6. The net income effect of the rural land rental market.**

| Matching methods | Welfare indicator | Renting-out welfare effect | | | | Renting-in welfare effect | | | |
|---|---|---|---|---|---|---|---|---|---|
| | | ATT[a] | Sensitivity test[a] | | | ATT[b] | Sensitivity test[b] | | |
| | | | $\Gamma = 1.5$ | $\Gamma = 1.0$ | $\Gamma = 0.5$ | | $\Gamma = 1.5$ | $\Gamma = 1.0$ | $\Gamma = 0.5$ |
| neighbor mat(1–5) | Family per capital net income | -632.5 | 0.065 | 0.071 | 0.053 | 25080.69* | 0.02 | 0.06 | 0.05 |
| neighbor mat(1–10) | | -144.79* | 0.056 | 0.061 | 0.052 | 25788.68** | 0.02 | 0.06 | 0.04 |
| kernel mat | | -451.16* | 0.048 | 0.049 | 0.051 | 24266 | 0.04 | 0.04 | 0.05 |
| radius mat | | 1040.73* | 0.053 | 0.049 | 0.052 | 30410.77* | 0.05 | 0.05 | 0.04 |
| Mean | | -567.3 | | | | 26387 | | | |

Note:

a and b denote the welfare effect of renting-out (renting-in) and non-participants, respectively.

land reform by the government brings more support (e.g., agricultural) for the renting-in rural households to realize scale-operation of farmlands. The focus is on enhanced agricultural productivity (Carter and Yao, 2002; Jin and Deininger 2009; Kimura et al., 2011) [18–20]. On the other hand, since our data only covers the initial third year of the land reform, the lag effect of the land system reform may decline the income for land renting-out rural households.

To encourage more participation in the land market, policymakers can provide more training to young farmers to cultivate the "professional farmers" who can understand technology. Improved understanding of agricultural technologies will enhance the land managers' ability to operate and manage large land areas and as well as reduce the potential loss caused by blind investment in land. The cultivation of more "professional farmers" can boost the land transaction activities in land markets and improve the productivity of land because skilled farmers are more likely to rent in the land. Other than providing technical training to farmers, policymakers can also use subsidies and provide more off-farm job opportunities to encourage more rural farmers to rent in and rent out the land. More land transaction activities in the rural area would, in the long run, help increase the productivity of farmland because skilled farmers are more likely to rent in land for scale farming. Meanwhile, unskilled farmers are more likely to rent out the land for other income opportunities.

We should interpret some results of this study with caution. Our results show that the income level of renting-out households is lower than those who do not participate in the land rental market. However, it does not mean that China's land reform is not successful. The income of renting-out households may, in the first place, be lower than those who do not rent out their land. Land reform potentially frees farmers from agriculture to find off-farm jobs that can earn them higher income. During the early years of the land reform, increased transaction costs may be the short-term frictional institutional cost. However, the transaction cost would be reduced or eliminated when most land in the market has been rent in or rent out. The limitation of the current study is that our data are only from the land reform demonstration zones. Subsequent studies can expand the sample areas with different levels of economic development and carry out research based on larger sample data. With more data collected, the analyses can be conducted for farmers with different land sizes, which can provide a deep insight into the effect of farmland rental markets on the income of households with various farm sizes.

## Supporting information

**S1 File.**
(XLSX)

## Author Contributions

**Data curation:** Chuchu Sun.

**Investigation:** Mengyao Wang.

**Writing – original draft:** Ning Geng.

**Writing – review & editing:** Zhifeng Gao.

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
