## [Decision Letter · Decision Letter 0]

2 Mar 2021

PONE-D-20-38442

How do farmland rental markets affect farmers’ income? Evidence from a matched renting-in and renting-out households survey in Northeast China

PLOS ONE

Dear Dr. Zhifeng Gao,

Thank you for submitting your manuscript to PLOS ONE. After careful consideration, we feel that it has merit but does not fully meet PLOS ONE’s publication criteria as it currently stands. Therefore, we invite you to submit a revised version of the manuscript that addresses the points raised during the review process.

Please submit a point by point rsponse to each comment indicating also where and how you address them in your revised manuscript.  Also we will require a copy  of the dataset used to produce the results of the paper.

We look forward to receiving your revised manuscript.

Kind regards,

Tahirou Abdoulaye

Academic Editor

PLOS ONE

Additional Editor Comments (if provided):

Dear Dr. Zhifeng Gao,

Based ont he reviews received, I would to invite you to revise and resubmit this manuscript. Please pay a lot of attention to reviewer 1 as I agree with most of this concerns.

Journal Requirements:

2) Please include additional information regarding the survey or questionnaire used in the study and ensure that you have provided sufficient details that others could replicate the analyses. For instance, if you developed a questionnaire as part of this study and it is not under a copyright more restrictive than CC-BY, please include a copy, in both the original language and English, as Supporting Information.

3) Thank you for stating the following financial disclosure:

 [YES - Specify the role(s) played.].

4)  We note that you have indicated that data from this study are available upon request. PLOS only allows data to be available upon request if there are legal or ethical restrictions on sharing data publicly. For more information on unacceptable data access restrictions, please see http://journals.plos.org/plosone/s/data-availability#loc-unacceptable-data-access-restrictions.

Reviewers' comments:

Reviewer's Responses to Questions

**Comments to the Author**

1. Is the manuscript technically sound, and do the data support the conclusions?

Reviewer #1: Yes

Reviewer #2: Yes

2. Has the statistical analysis been performed appropriately and rigorously? 

Reviewer #1: Yes

Reviewer #2: Yes

3. Have the authors made all data underlying the findings in their manuscript fully available?

Reviewer #1: Yes

Reviewer #2: No

4. Is the manuscript presented in an intelligible fashion and written in standard English?

Reviewer #1: Yes

Reviewer #2: Yes

5. Review Comments to the Author

Reviewer #1: The study used a farm household survey of 810 rural households in China to examine the factors influencing participation in the farmland rental market. The study also evaluated the impact of the rental market on farmers’ income. Results of the study show that household off-farm income, family members’ part-time employment, and agricultural subsidies, agricultural cooperatives significantly affect farmers’ participation in the farmland rental market. Finally, participation in the farmland rental market increases the income of renting-out households.

Generally, the subject of investigation is relevant within the context of land rental market. I find the research interesting especially within the context of developing countries where the issues of the welfare effect of renting-out households in the rental market are often ignored. The methodology employed for the study is relevant and adequately addressed the research questions. The conclusion of the study is adequately supported by the findings. However, the paper will improve significantly if the major and minor comments are adequately addressed. I recommend a MAJOR revision.

Reviewer #2: This paper addresses an important question: How do farmland rental markets affect farmers’ income? Evidence from a matched renting-in and renting-out households survey in Northeast China. The background to the study shows that farmland market renting is a very important issue in China where land is not owned by individual farmers. The authors present and discuss a good conceptual framework that leads to the use of a logit model to estimate the determinants of the rental market and the PSM models to measure the effect of the renting market on household income. The argument presented by authors to use this methodology is that it accounts for self-selection bias. They used data covering 810 rural households across 22 villages from Shandong province, which to them can be considered including a representative sample of participants of the rural land rental market in the targeted region.

The authors’ findings indicate that household off-farm income, family members’ part-time employment, and agricultural subsidies, agricultural cooperatives significantly affect farmers’ participation in the farmland rental market. Also, participation in the farmland rental market significantly increases the income of renting-in households, while it decreases the income of renting-out households which might result from the lag effect of the land system reform temporarily.

The flow of the paper making easy to read.

However, there are concerns that if addressed could improve the quality of the paper:

Data: The authors provided the reason why the two cities of Shandong province Dezhou and Qingdao were selected. However, the reasons why the selection of townships and villages differ between the two provinces are not given. Besides, nothing is said on how individual respondents were selected at the village level. There is the need to provide more light on the rationale in the distribution of the sample across participating and non-participating households on the one side, and across provinces and villages on the other side. Much more information is needed on the dataset – ie when the data were collected (ie, when during the agricultural season). Besides, describing how the outcome variables (income in this case and other key variables) were empirically measured is helpful particularly. We know that farmers in most of the cases have measurement problems, what effort was undertaken to reduce the measurement error on the outcome’s variables?

Participation: The definition of participation as given by the authors is clear enough. However, the definition is silent about the time frame that a household should participate to be considered as “Participant”. This is important for attribution.

Endogeneity. The current methodology used focuses on the selectivity bias. However, the authors raised the issue of bias due to unobservable factors. Some tests need to be performed to clarify whether or not these factors create the problem of endogeneity and if yes propose a way to deal with the problem.

Discussion and policy relevance: How do these results fit into the existing literature in this area? There was no effort by the authors to compare the results obtained in the study with the one in the literature. What is the policy relevance of this research? The current analysis supposes that participating in farmland renting market affect household the same way which is not the case in practice. It would be good to apply some decomposition perhaps by some socioeconomics characteristics such as gender to appreciate that difference.

Finally, the paper would benefit from some additional editing to ensure that concepts and terminology are stated clearly. Some of my edits are in the attachment.

6. PLOS authors have the option to publish the peer review history of their article (what does this mean?). If published, this will include your full peer review and any attached files.

Reviewer #1: No

Reviewer #2: No

---

## [Author Response · Author response to Decision Letter 0]

26 May 2021

Please see attached file for our replies to reviewers` comments.

Reviewer #1

Reviewer #1: The study used a farm household survey of 810 rural households in China to examine the factors influencing participation in the farmland rental market. The study also evaluated the impact of the rental market on farmers’ income. Results of the study show that household off-farm income, family members’ part-time employment, and agricultural subsidies, agricultural cooperatives significantly affect farmers’ participation in the farmland rental market. Finally, participation in the farmland rental market increases the income of renting-out households.

Generally, the subject of investigation is relevant within the context of land rental market. I find the research interesting especially within the context of developing countries where the issues of the welfare effect of renting-out households in the rental market are often ignored. The methodology employed for the study is relevant and adequately addressed the research questions. The conclusion of the study is adequately supported by the findings. However, the paper will improve significantly if the major and minor comments are adequately addressed. I recommend a MAJOR revision.

Reply:

Thank you for the comments. We have addressed the issues that you raised below item by item. We have added new figures, running some further analyses to address your concerns. In the places where no revisions are done, we carefully explained what we have done and why to choose what we have done. We hope the revised manuscript adequately addressed all your concerns. 

Major Comments

1. The author(s) should show the graph indicating the region of common support.

Reply: Thank you for your suggestion. We added two new graphs in the revised manuscript titled “Fig.4. After matching of renting-out households” and “Fig.5. After matching of renting-in households” to indicate the region of common support of land transfer households (renting-out households and renting-in households) and non-participants (Page 16). 

2. In the Rosenbaum sensitivity test you conducted; it will be appropriate to show the performance of the estimates at the different levels of gamma values rather than showing only the gamma value of 1.5.

Reply: Thank you for your suggestion. We have added two columns to the table to show the p-values at different levels of gamma values (1.0 and 0.5) in the revised manuscript (Page 19 Table 6). The results show that our conclusions are robust. 

Table 6. The net income effect of rural land rental market 

Matching methods Welfare indicator Renting-out welfare effect Renting-in welfare effect

 ATT a Sensitivity test a ATT b Sensitivity test b

 Γ=1.5 Γ=1.0 Γ=0.5 Γ=1.5 Γ=1.0 Γ=0.5

neighbor mat （1-5） Family per capita net income -632.46 0.065 0.071 0.053 25080.69� 0.02 0.06 0.05

neighbor mat -144.79� 0.056 0.061 0.052 25788.68�� 0.02 0.06 0.04

(1-10) 

kernel mat -451.16� 0.048 0.049 0.051 24265.97 0.04 0.04 0.05

radius mat -1040.73� 0.053 0.049 0.052 30410.77� 0.05 0.05 0.04

Mean -567.29 26386.53 

Notes: a denotes the welfare effect of renting-out and non-participants; b denotes the welfare effect of renting-in and non-participants

3.The discussion of the results must be strengthened.

Reply: Thank you for your suggestion. We strengthened the discussion of the results by adding more discussion in the conclusion section as well as providing more explanations of our results. 

First, related to the results of determinants of farmers' participation in the land rental market, we strengthened the analysis of some explanatory variables. For instance, we explain more about the coefficients of “share” and “parttime” in Table 4 that denote household members' off-farm employment. We added, “due to the low comparative returns of agriculture, the non-agricultural production activities of ordinary farmers are gradually transferred to the primary service industries in the surrounding areas and urban areas.” (Page 18)

For other explanatory variables such as “land price” and “cooperative”, we added, “In line with our theoretical analysis, land price is a key element of the land rental market. The higher the land rent would increase the cost of land for production, therefore having a negative effect on farmers' rent-in behavior.” We also added, “ Besides, the control variable "cooper" has a significant impact on the renting-out land at 1% significance level. This implies joining cooperatives may reduce transaction costs of agricultural production or marketing, therefore resulting in a lower probability of land rent-out behavior.” (page 19) 

Secondly, we strengthened the conclusion by adding more discussion and comparing our results with previous research. We added, “Moreover, we also need to focus on cultivating "professional farmers" who are able to understand technology. The level of education and mastery of land managers' agricultural technology determine their ability to operate and manage land. Therefore, improving land managers' skills can reduce the risk of investment in land for agricultural production. ” (page 19)

4.Differential analysis - an additional suggested analysis could be categorizing the farmers (rent-in and rent-out) into different land size (terciles) and perform the PSM. This will allow the author(s) to establish the sensitivity of their results based on changes in the land size.

Reply: Thank you for your suggestion. We have tried to classify farmers (rent-in and rent-out) by their land size (0-30 and 30-50 or 0-25, 25-50, 50-100, 100 above) and performed PSM. However, after categorizing the farmers, the sample size of each sample became too small to run PSM analysis because large sample sizes are required for PSM analysis. 

To address this concern, we included your suggestion as a future research direction at the end of the conclusion. We added, “Finally, with more data collected, similar analyses can be conducted for farmers with different land sizes, which can provide a deep insight into the effect of farmland rental markets on the income of various farm sizes.” (page 25)

Minor Comments

1.There are some grammatical errors and spelling mistakes that might have escaped the attention of the author(s) – For example page 15, 16, 17, 25

Reply: Thank you for your comment. We have carefully checked the paper and corrected grammatical errors and spelling mistakes.

2.The mathematical notations of the variables must be clear and precise.

Reply: Thank you for your comment. We rectified the notation for the variables again. 

3.Use formal writing style. E.g. use “cannot” instead of “can’t”.

Reply: Thank you for your comment. We have made the changes. 

4.The tables must be properly formatted. 

Reply: Thank you for your comment. We have formatted the tables as required by PLoS ONE guidline.

5.I recommend a thorough formatting of the equations and the text as well.

Reply: Thank you for your comment. We have performed thorough formatting of the equations and the text in the revised manuscript.

Reviewer #2: 

This paper addresses an important question: How do farmland rental markets affect farmers’ income? Evidence from a matched renting-in and renting-out households survey in Northeast China. The background to the study shows that farmland market renting is a very important issue in China where land is not owned by individual farmers. The authors present and discuss a good conceptual framework that leads to the use of a logit model to estimate the determinants of the rental market and the PSM models to measure the effect of the renting market on household income. The argument presented by authors to use this methodology is that it accounts for self-selection bias. They used data covering 810 rural households across 22 villages from Shandong province, which to them can be considered including a representative sample of participants of the rural land rental market in the targeted region.

The authors’ findings indicate that household off-farm income, family members’ part-time employment, and agricultural subsidies, agricultural cooperatives significantly affect farmers’ participation in the farmland rental market. Also, participation in the farmland rental market significantly increases the income of renting-in households, while it decreases the income of renting-out households which might result from the lag effect of the land system reform temporarily.The flow of the paper making easy to read.

Reply:

Thank you for the comments. We have addressed the issues that you raised below item by item. We have added new figures, running some new analyses to address your concerns. In the places where no revisions are done, we carefully explained what we have done and why to choose what we have done. We hope the revised manuscript adequately addressed all your concerns. 

However, there are concerns that if addressed could improve the quality of the paper:

Data: The authors provided the reason why the two cities of Shandong province Dezhou and Qingdao were selected. However, the reasons why the selection of townships and villages differ between the two provinces are not given. Besides, nothing is said on how individual respondents were selected at the village level. There is the need to provide more light on the rationale in the distribution of the sample across participating and non-participating households on the one side, and across provinces and villages on the other side. Much more information is needed on the dataset – ie when the data were collected (ie, when during the agricultural season). Besides, describing how the outcome variables (income in this case and other key variables) were empirically measured is helpful particularly. We know that farmers in most of the cases have measurement problems, what effort was undertaken to reduce the measurement error on the outcome’s variables? 

Reply: Thanks for the comment. Please see our reply to your comment below. 

First, regarding the comments “reasons why the selection of townships and villages differ between the two provinces are not given.”

“Shandong is one of the most densely populated provinces of China, with small farms and high levels of land fragmentation cover approximately 157,100 km2. As the major grain-producing region, Shandong's grain production in 2017 was 53.2 million tons (Fig 2), accounting for 8 percent of the national total. Shandong led the country in total grain output value in 2017 (Fig 3), reaching $60.635 billion.” (Page 8). We select Dezhou and Qingdao of Shandong province, “because they are agricultural demonstration zones in Shandong, and more than 40% of farmland is traded in the land market.” And one of the key difference between the two cities is that “the proportion of traded land in Qingdao is approximate 50%, while Dezhou trades 42%.”(Page 8)

Second, regarding “nothing is said on how individual respondents were selected at the village level”, 

We added, “we randomly selected respondents in the sample villages” in the revised manuscript (page 11).

Third, regarding the comment “There is the need to provide more light on the rationale in the distribution of the sample across participating and non-participating households on the one side, and across provinces and villages on the other side.” 

Among the total 810 rural households interviewed, there are 407 participating households, including 240 renting-out households and 167 renting-in households. The specific distribution of these households by the city is that there are 559 rural households interviewed in Qingdao, with 266 participating households that include 165 renting-out households and 101 renting-in households. In comparison, there are 251 rural households interviewed in Dezhou, with 141 participating households that include 75 renting-out households and 66 renting-in households. The samples are distributed in such a way mainly because the population in Qingdao is much larger than that of Dezhou. In the manuscript, we did not provide detailed distribution of participating vs. non-participating households by city because our study does not try to determine the welfare effect of land market participation by city. In addition, it is not possible to conduct PSM analysis if we separate the sample by city because of the smaller sample size for each city. 

Fourth, regarding the comment that “Much more information is needed on the dataset – ie when the data were collected (ie, when during the agricultural season).” 

We randomly selected respondents in the sample villages during the non-farming season because most farmers are busy with agricultural production and do not have much time to participate in the survey during the farming season. On page 10, we stated that “The data used in this article were collected from a socio-economic survey of landholders from December 2017 to November 2018 during the non-farming season in Shandong.” 

At last, regarding the comment that “Besides, describing how the outcome variables (income in this case and other key variables) were empirically measured is helpful particularly. We know that farmers in most of the cases have measurement problems, what effort was undertaken to reduce the measurement error on the outcome’s variables?” 

In our study, income is measured by asking the interviewee “household incomes per capita ”. Several methods are used to reduce the measurement errors of the outcome variable. For instance, face-to-face interviews were used to increase the trust between the interviewer and interviewee, therefore increasing the response rate and obtain respondents' truthful opinions. Before the survey was carried out, we also made it clear to the interviewees in advance that our survey was only for scientific research and had no political purpose.

In the revised manuscript, we have 

“Finally, to increase the response rate and obtain respondents' truthful opinions, we conducted the survey by face-to-face interviews. Before the survey was carried out, we also made it clear to the interviewees in advance that our survey was only for scientific research and had no political purpose. During the field survey, one enumerator took 1-2 hours to complete one questionnaire and each survey enumerator completed 3-5 questionnaires per day. To ensure the authenticity of the survey，we randomly selected respondents in the sample villages. Because our work focused on the land renting-in and renting-out behavior, when we interviewed the renting-out households, we should also find out the matching renting-in household to interview. To accurately estimate the income effect of the land rental market, we also interviewed farmers who were not participating in the land rental market in the sample villages.” (page 10-11 )

Participation: The definition of participation as given by the authors is clear enough. However, the definition is silent about the time frame that a household should participate to be considered as “Participant”. This is important for attribution.

Reply: Thanks for the comments. In our survey, the time frame of participants was determined through lease contracts. Most of the contract was between 8-20 years. However, the current study mainly focuses on the welfare effect of participation in the land market rather than the effect of contract length. Future research can investigate the impact of land rent-in/rent-out contract length on household welfare. 

In the revised manuscript, we stated “Finally, most of the land lease agreement was between 8-20 years. The current study mainly focuses on the welfare effect of participation in the land market rather than the effect of lease agreement length. Future research can investigate the impact of land rent-in/rent-out contract length on household welfare.” (page 11)

Endogeneity. The current methodology used focuses on the selectivity bias. However, the authors raised the issue of bias due to unobservable factors. Some tests need to be performed to clarify whether or not these factors create the problem of endogeneity and if yes propose a way to deal with the problem.

Reply: Thanks for the comment. The main reason that we used PSM in the current research is to avoid the endogeneity issues that may be associated with regression analysis. Thus, we are not quite sure what you mean by “these factors create the problem of endogeneity.” If you can provide any reference for us to check, it would be really appreciated. 

PSM may be sensitive to “unobservable factors” that are not included in the covariates when we did the matching. To make sure our results are robust and will not be affected by the unobservable factors, we conducted a sensitivity analysis and reported additional results in Table 6. Table 6 added two columns to show the p-values at different levels of gamma values (1.0 and 0.5). The results show that our conclusions are insensitive to the unobservable factors, therefore robust. 

Discussion and policy relevance: How do these results fit into the existing literature in this area? There was no effort by the authors to compare the results obtained in the study with the one in the literature. What is the policy relevance of this research? The current analysis supposes that participating in farmland renting market affect household the same way which is not the case in practice. It would be good to apply some decomposition perhaps by some socioeconomics characteristics such as gender to appreciate that difference.

Reply: we separate your comments and will reply to them one by one. 

1. How do these results fit into the existing literature in this area? There was no effort by the authors to compare the results obtained in the study with the one in the literature.

Reply: First, in the introduction, the authors the existing literature mainly focused on efficiency, equity, and welfare. The impact on welfare, and the existing literature is discussed separately from renting out households and renting in households. Second, to add more discussion of past research in the results and discussion section, we cite more literatures to fit these results.( Page 18 and 19)

2. What is the policy relevance of this research? The current analysis supposes that participating in farmland renting market affect household the same way which is not the case in practice.

Reply: This research wants to solve the question about how the rental market improve income in the smallholder farming system for both renting-in and renting-out household. Our results show that for renting-in households, joining the rural land rental market can increase farmers' welfare, which is consistent with the previous research conclusions(Fei Chen,2015; Min Shi ,2017). While from the perspective of renting-out households, joining the rural land rental market reduces the welfare of farmers, which is inconsistent with the previous research conclusion. This may be related to the lagging effect of land system reform ( Lan Z. Shuyi F. ,2017) . So our research does not assume that “participating in farmland renting market affect household the same way”. And our research is an extension of the existing literature in this area. So policy relevance is, Land reforms have greatly stimulated the development of China's rural land rental market, especially participation in the farmland rental market significantly increase the income of renting-in households. Due to the lagging effect of land reform, the increase in the income of renting-out households has not been obvious, so we should provide more off-farm jobs to renting-out households while cultivate more "professional farmers" who are able to understand agricultural technology.

3. It would be good to apply some decomposition perhaps by some socioeconomics characteristics such as gender to appreciate that difference

Thanks for the comments. This is a good suggestion. However, we could not separate the samples by gender and do the PSM to see whether the welfare of participation differs by gender or not. The reason is that we required the interviewee to be a person who is familiar with agricultural production. Although both husband and wife in rural China make joint decisions most of the time, males are mainly responsible for agricultural production and thefore are more familiar with the land rental issues. For instance, males account for 97% of the sample in this study. Therefore, it is hard to divide the samples by gender. With more samples collected, we could try to differentiate the welfare effect of the farmland renting market participation by some demographics or farm charactristics (e.g., farm land size). We added this as a future research directions at the end of the paper. 

“Subsequent studies will expand the sample areas with different levels of economic development and carry out research based on larger sample data. With more data collected, similar analyses can be conducted for farmers with varying sizes of land or demographics, which can provide a deep insight into farmland rental markets' welfare effect.” (page 25)

Finally, the paper would benefit from some additional editing to ensure that concepts and terminology are stated clearly. Some of my edits are in the attachment.

Reply: We appreciate your edits and comments. We also go through the paper thoroughly to make sure that all the concepts and terminologies are stated clearly.

---

## [Decision Letter · Decision Letter 1]

11 Aug 2021

How do farmland rental markets affect farmers’ income? Evidence from a matched renting-in and renting-out households survey in Northeast China

PONE-D-20-38442R1

Dear Dr. Ning Geng

We’re pleased to inform you that your manuscript has been judged scientifically suitable for publication and will be formally accepted for publication once it meets all outstanding technical requirements.

Kind regards,

Tahirou Abdoulaye

Academic Editor

PLOS ONE

Additional Editor Comments (optional):

thank you for choosing Plos One. You have addressed the concerned of our 2 reviewers I am therefore please to accept your paper for publication.

Reviewers' comments:

Reviewer's Responses to Questions

**Comments to the Author**

1. If the authors have adequately addressed your comments raised in a previous round of review and you feel that this manuscript is now acceptable for publication, you may indicate that here to bypass the “Comments to the Author” section, enter your conflict of interest statement in the “Confidential to Editor” section, and submit your "Accept" recommendation.

Reviewer #1: All comments have been addressed

Reviewer #2: All comments have been addressed

2. Is the manuscript technically sound, and do the data support the conclusions?

Reviewer #1: Yes

Reviewer #2: Yes

3. Has the statistical analysis been performed appropriately and rigorously? 

Reviewer #1: Yes

Reviewer #2: Yes

4. Have the authors made all data underlying the findings in their manuscript fully available?

Reviewer #1: No

Reviewer #2: Yes

5. Is the manuscript presented in an intelligible fashion and written in standard English?

Reviewer #1: Yes

Reviewer #2: Yes

6. Review Comments to the Author

Reviewer #1: All the comments have been adequately addressed to my satisfaction. However, the introduction needs to be reduced and the research question introduced early in the introduction.

Reviewer #2: (No Response)

7. PLOS authors have the option to publish the peer review history of their article (what does this mean?). If published, this will include your full peer review and any attached files.

Reviewer #1: **Yes: **Edward Martey

Reviewer #2: No

---

## [Editor Report · Acceptance letter]

21 Sep 2021

PONE-D-20-38442R1 

How do farmland rental markets affect farmers’ income? Evidence from a matched renting-in and renting-out household survey in Northeast China 

Dear Dr. Geng:

I'm pleased to inform you that your manuscript has been deemed suitable for publication in PLOS ONE. Congratulations! Your manuscript is now with our production department. 

Kind regards, 

on behalf of

Dr. Tahirou Abdoulaye 

Academic Editor

PLOS ONE